

# Air Quality Predictions with an Analog Ensemble

Luca Delle Monache[1], Stefano Alessandrini[1], Irina Djalalova[2], James Wilczak[2], Jason C. Knievel[1]

[1]National Center for Atmospheric Research[*], P.O. Box 3000, Boulder, Colorado, USA, 80307-3000
[2]National Oceanic and Atmospheric Administration, 325 Broadway, Boulder, Colorado, USA, 80305-
3337

*Correspondence to*: Dr. Luca Delle Monache (lucadm@ucar.edu)

**Abstract.**   The authors demonstrate how the analog ensemble (AnEn) can efficiently generate

deterministic and probabilistic forecasts of air quality.   AnEn estimates the probability of future

observations of a predictand based on a current deterministic numerical weather prediction and an archive

of prior analog predictions paired with prior observations.   The method avoids the complexity and real-

time computational expense of dynamical (i.e., model-based) ensembles.   The authors apply AnEn to

observations from the Environmental Protection Agency's (EPA's) AIRNow network and to forecasts

from the Community Multiscale Air Quality (CMAQ).   Compared to raw forecasts from CMAQ,

deterministic forecasts of $O_3$ and $PM_{2.5}$ based on AnEn's mean have lower errors, both systemic and

random, and are better correlated with observations.   Probabilistic forecasts from AnEn are statistically

consistent, reliable, and sharp, and they quantify the uncertainty of the underlying prediction.

---

[*] The National Center for Atmospheric Research is sponsored by the National Science Foundation



# 1 Introduction

Every year poor air quality kills millions of people worldwide (Forouzanfar et al., 2015) and in the U.S. alone costs society tens to hundreds of $billions (Muller and Mendelsohn, 2007). Air-quality forecasts are one resource that decision-makers can use to reduce many threats that poor air quality poses.

5        The lower the uncertainty of the information on which decisions are based, the better. Unfortunately, uncertainty cannot be completely eliminated from air-quality forecasting, but there are effective ways to treat the inevitable uncertainty and to reduce its consequences. One way is through probabilistic ensemble prediction. In contrast to the *deterministic* approach of using a single forecast from a single model, an ensemble comprises multiple, meaningfully different forecasts that are valid at

10  the same future time and location, from which *probabilistic* information can be obtained.

       Ensembles are beneficial in many ways. The probabilistic guidance they provide is potentially much more useful for decision-makers than a single forecast could ever be (Buizza, 2008; Palmer, 2002). An ensemble's mean forecast tends to be (but is not always) more skillful than any individual member's prediction (Delle Monache et al., 2006a, 2006b, 2008; Delle Monache and Stull, 2003; Djalalova et al.,

15  2010; Du et al., 1997; Ebert, 2001; Galmarini et al., 2001; Galmarini et al. 2004; Kioutsioukis and Galmarini et al. 2014; Leith, 1974; McKeen et al., 2005; Potempski et al. 2008; Potempski and Galmarini, 2009; Solazzo et al. 2012; Toth and Kalnay, 1997; Zhu et al., 2012). Calculating the mean filters out some of the unpredictable elements of the physical processes being simulated. Another benefit is that the approximate uncertainty in a mean forecast can be inferred from the spread among ensemble members

20  (Kalnay, 2003) if the ensemble is calibrated, although that inference has to be made carefully and is not



valid in every case (Barker, 1991; Hopson, 2014; Murphy, 1988). Ensembles also produce *reliable* and

*well resolved* probabilistic forecasts that can be further improved through calibration and other methods

of post-processing imperfect numerical predictions. A *reliable* ensemble is one that over many cases

predicts conditions to occur with the same frequency as they actually occur in nature. A *well resolved*

5   ensemble is one that is specific from case to case about what conditions will and will not occur.

In air-quality forecasting in particular, probabilistic approaches have been recommended as—and

have been demonstrated to be—effective at dealing with the many sources of uncertainty (e.g., Bei et al.,

2010; Carmichael et al., 2008; Dabberdt et al., 2004; Delle Monache et al., 2006a, 2006b, 2008; Delle

Monache and Stull, 2003; Garaud and Mallet, 2010; Kioutsioukis et al., 2016; Marécal et al., 2015; Mallet,

10   2010; Mallet et al., 2013; Mallet and Sportisse, 2006a, 2006b; McKeen et al., 2005; Pagowski et al., 2005;

Zhang et al., 2007; Zhang et al., 2012). Uncertainties stem from meteorological initial and boundary

conditions; errors in observations assimilated into a model; truncations and approximations in a model's

numerical schemes; specification of emissions, which are often not well known nor well characterized in

models; and physical or chemical processes that are simplified or omitted entirely (Delle Monache and

15   Stull, 2003).

Ensemble prediction takes different forms. One of the simplest is a lagged ensemble Dalcher et

al., 1988; Delle Monache et al., 2006a; Ebisuzaki and Kalnay, 1991; (Hoffman and Kalnay, 1983; Lu et

al., 2007; Mittermaier, 2007). In a lagged ensemble, sequential forecasts from a deterministic system are

grouped together, each with a common valid time but with different lead times (e.g., an 18-h forecast

20   initialized at 0000 UTC and valid at 1800 UTC, a 15-h forecast initialized at 0300 UTC and valid at 1800





UTC, and so on). The effectiveness of this approach depends on how frequently the deterministic forecasts are updated (Lu et al., 2007; Mittermaier, 2007).

Lagged ensembles target uncertainty in the initial conditions. So do other, similar, single-model ensembles in which different initial conditions, boundary conditions, and/or perturbed observations are

5 used to generate diversity even though the model code (including physical parameterizations) is fixed (e.g., Molteni et al., 1996; Zhang et al., 2007). In modelling air quality, perturbations can be applied not just to meteorological observations, but also to emissions. Considering both types of perturbations is important for addressing the nonlinearity of the problem (Delle Monache et al., 2006a, 2006c).

More complex and computationally expensive—and often more effective—are ensembles based

on multiple models. Multi-model ensembles address uncertainty in the actual physical or chemical processes being simulated, not merely in the initial meteorological or chemical conditions (e.g., Bei et al., 2010; Delle Monache et al., 2008; Delle Monache and Stull, 2003; Djalalova et al., 2010; Garaud and Kioutsioukis et al., 2016; Mallet, 2010; Mallet and Sportisse, 2006a, 2006b; McKeen et al., 2005; Vautard et al., 2012). (For the purposes of this article, we consider what are sometimes called *multi-physics*

ensembles to be just one type of *multi-model* ensemble.) Multi-model ensembles have proved to be a particularly effective tool for probabilistic operational weather forecasting for quite some time (e.g., Buizza et al., 2005; Hacker et al., 2011; Krishnamurti, 1999). In air-quality forecasting, multi-model ensembles have been successfully applied to forecasts of ground-level ozone (e.g., McKeen et al., 2005; Solazzo et al., 2012; Žabkar et al., 2013), airborne particles (e.g., Djalalova et al. 2010; McKeen et al.,

2007), and to both (e.g., Monteiro et al., 2013).

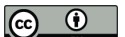



The complexity and expense of multi-model ensembles and the limitations of single-model ensembles are two motivations for developing simpler, more cost-effective, yet still powerful methods for probabilistic prediction—especially hybrid statistical-dynamical methods. The goal of this article is to describe how one such method, an analog ensemble (AnEn), can be applied to air-quality forecasting.

5  We compare AnEn's forecasts of ground-level ozone ($O_3$) and particles less than 2.5 µm in diameter ($PM_{2.5}$) to the simpler probabilistic standard of a persistence ensemble (PeEn), and to an operational standard of the industry, the Environmental Protection Agency's (EPA's) Community Multiscale Air Quality (CMAQ) model (Byun and Schere, 2006). The two ensemble methods, CMAQ, and the observations used for this research are described in the following sections.

## 2  Predictive systems and data

### 2.1  Analog ensemble (AnEn)

AnEn is a hybrid statistical-dynamical method to generate an ensemble for estimating the probability of some future observation of a predictand (e.g., 2-m dew point, geopotential height at 500 hPa, or, in the case of air-quality forecasts, $PM_{2.5}$) given a current deterministic prediction and an archive of historical,

15  analogous deterministic predictions paired with historical observations at those predictions' valid times. That archive is used to train the AnEn. Typically, the deterministic predictions in the training data and the current deterministic prediction come from the same configuration of the same NWP model, or nearly so. Hamill and Whitaker (2006) demonstrated the considerable value of using an analog ensemble approach to calibrate an existing ensemble, and Delle Monache et al. (2013) proposed a similar approach

20  to instead generate an ensemble, which form the basis of our approach.



If the future observation of a predictand is represented by $y$, then the probability distribution of

that future observation is

$$f(y|\boldsymbol{x}^f),$$

wherein $\boldsymbol{x}^f = \left(x^f_{1,}\, x^f_{2,}\, x^f_{3,}\, \ldots\, x^f_{k,}\right)$ are the $k$ predictors from the deterministic model forecast. How an

analog is defined, how its quality is assessed, how many analogs to select from the archive, how deep an

archive is necessary, etc., can vary from one application of the AnEn to another. Section 3 goes into more

detail about our application's sensitivity to some of those choices.

For determining which historical forecasts are sufficiently analogous to the current forecast, we

follow Delle Monache et al. (2011, 2013) and Nagarajan et al. (2015) in using the metric developed by

Delle Monache et al. (2011):

$$\left\| F_t, A_{t'} \right\| = \sum_{i=1}^{N_v} \frac{w_i}{\sigma_{f_i}} \sqrt{\sum_{j=-\tilde{t}}^{\tilde{t}} \left( F_{i,t+j} - A_{i,t'+j} \right)^2},$$

wherein $F_t$ is the current deterministic forecast for a location of interest, valid at some time $t$ in the future;

$A_{t'}$ is an analogous forecast from the archive, valid at some time $t'$ in the past, and with the same lead

time as the current forecast's; $N_v$ is the number of atmospheric variables, i.e., predictors, used to select

analogs; $w_i$ is the weight assigned to each atmospheric variable of index $i$; and $\sigma_{f_i}$ is the standard

deviation of historical forecasts of each atmospheric variable of index $i$. The metric is calculated over a

range of times from $-\tilde{t}$ to $+\tilde{t}$ centered on the valid time $t$. $F_{i,t+j}$ and $A_{i,t'+j}$ respectively are each current

forecast and each analogous historical forecast for atmospheric variable index $i$ within that range of times.

We set $\tilde{t} = 1$. The forecast interval is 0–48 h.



The weights $w_i$ for each analog predictor were determined independently for each observing site as explained in section 3.2. From each search of the archived datasets, the best 20 analogs were chosen, which for any given search is 3–4% of the total cases in the archive. How many analogs to choose is a balance between sampling enough of the observed distribution of the predictand variable while ensuring

that all analogs are sufficiently similar to the current prediction.

## 2.2 Persistence ensemble

Like Alessandrini et al. (2015), we use a persistence ensemble (PeEn) as a baseline method for probabilistic prediction, then demonstrate AnEn's improvement on that baseline. For each forecast lead time, PeEn is based on the most recent 20 observations of $O_3$ or $PM_{2.5}$ at the same hour of day as the

forecast valid time. Other tested PeEn configurations (which do not perform as well as the latter, not shown) include PeEn formed by the most recent observations collected the previous seven days over a three-hour window centered on the same hour of the day, and a configuration including observations from the previous four days over a five-hour window. The PeEn ensemble can be skillful when air quality conditions persist for several days, or when conditions fluctuate with the same repeating diurnal pattern.

Rapidly changing patterns of $O_3$ or $PM_{2.5}$ are challenging for the PeEn.

## 2.3 CMAQ

CMAQ (Byun and Schere, 2006), with the Chemistry-Transport Model (CTM) it comprises, is an industry standard in air-quality forecasting. It is a modular, Eularian, Cartesian modeling system for simulating on regional scales the emission, production, advection, diffusion, chemical transformation, and removal

of atmospheric pollutants. As input into AnEn's algorithms, we use CMAQ's daily forecasts and ground-



level concentrations of $O_3$ and $PM_{2.5}$ at lead times of 0–48 hours. Additional inputs for AnEn include 10-m wind speed and direction, 2-m air temperature, 2-m specific humidity, and cloud cover, which were extracted from the numerical weather predictions that are used to drive CMAQ, the National Centers for Environmental Prediction (NCEP) North America Model (NAM).

5  **2.4 Observations**

The source of observations is the Environmental Protection Agency's (EPA's) AIRNow network (EPA, 2017) in the conterminous U.S. and southern Canada (Figure 1). Hourly concentrations of $O_3$ are from 1337 sites, of $PM_{2.5}$ from 551 sites. In this study, we implement a quality control procedure of the observations that is suitable for real-time operational forecasting. A detailed description of this procedure

10  applied to $PM_{2.5}$ can be found in section 2 of the paper by Djalalova et al. (2015).

Additionally, observation sites frequently reporting missing data are excluded by the analysis presented in this study, i.e., only stations with at least 50% of data available are retained. That results in 1045 and 458 sites for $O_3$ and $PM_{2.5}$, respectively, available to generate AnEn.

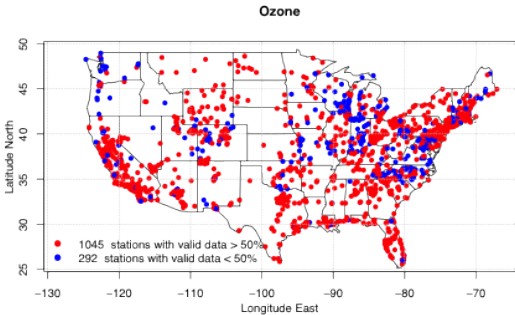
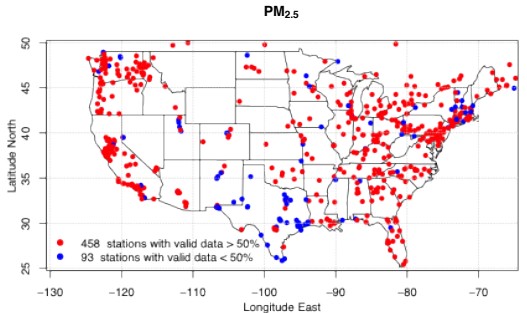





**Figure 1.** Sites of observations of a) $O_3$ and b) $PM_{2.5}$ used for this study. Colors indicate whether hourly observations at a site are available more (red) or less (blue) than 50% of the time during the two periods of study.

## 3 Results

This section starts with examples of AnEn's air-quality forecasts, followed by sensitivity tests of the

5 ensemble's algorithm as the number of analogs and the length of the training data are varied, and a

description of the analog predictor weights. Then we present an in-depth analysis of AnEn's performance

compared to CMAQ's (for deterministic predictions) and PeEn's (for probabilistic predictions). Our

periods of study are 1 July 2014 through 30 September 2015 (456 days) for predictions of $O_3$ and 1 July

2014 through 29 February 2016 (608 days) for predictions of $PM_{2.5}$. The verification periods were June

through September 2015 for $O_3$ and December 2015 through February 2016 for $PM_{2.5}$.

### 3.1 Examples of forecasts from AnEn, PeEn, and CMAQ

Figure 2 shows examples of $O_3$ (top) and $PM_{2.5}$ (bottom) predictions by the methods considered in this

study. For $O_3$, both ensemble means reduce some of the CMAQ biases, particularly at night when $O_3$

titration processes are a modelling challenge, leading CMAQ to overestimate $O_3$ concentration close to

15 zero. AnEn's mean is closer to the observations than is PeEn's. The usefulness of probabilistic

predictions is evinced for $O_3$ at forecast lead times 28–30. The deterministic predictions by CMAQ and

the ensemble means miss the observed peak, but the ensemble spread indicates from both AnEn and PeEn

a low probability of higher concentration, which could be useful information for a decision-maker trying

to protect the public from unhealthful air. Similar qualitative comments can be made about the $PM_{2.5}$

predictions shown in the bottom panels.





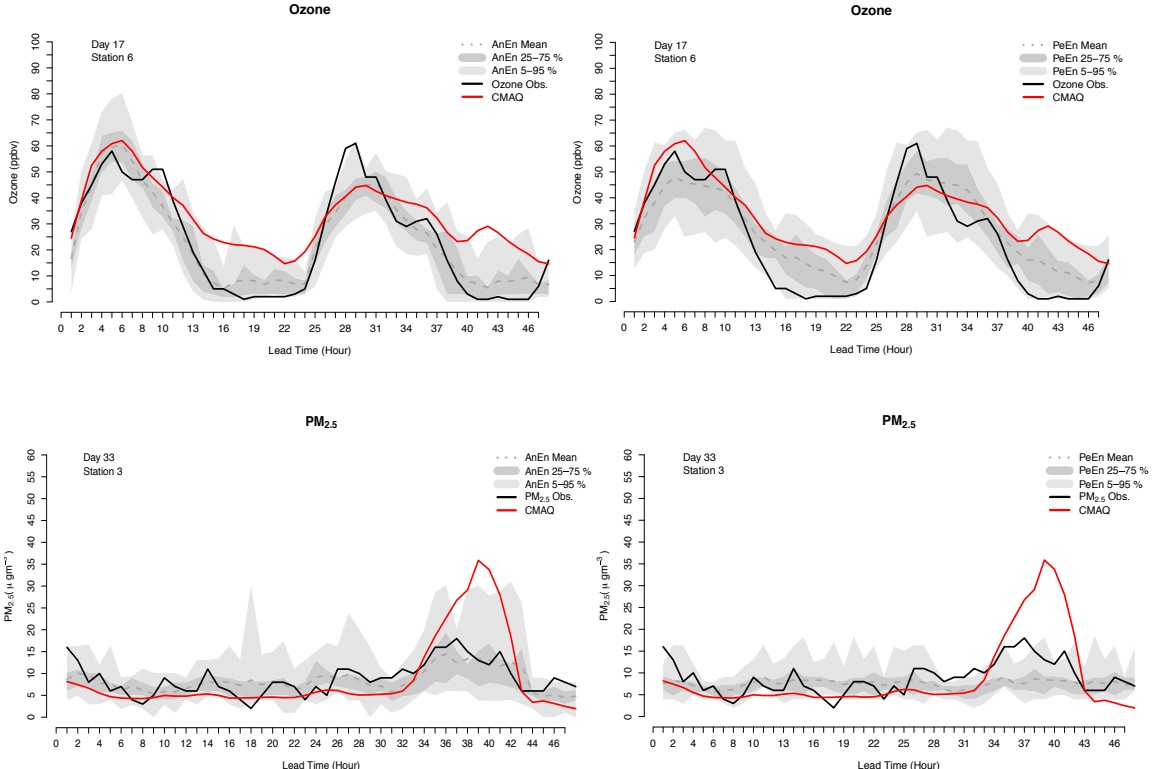

**Figure 2.** Examples of 0-48 predictions of $O_3$ (top) and $PM_{2.5}$ (bottom) at two different locations and days. AnEn's results are on the left, PeEn's on the right. CMAQ's predictions are in red, observations in black. Both ensemble distributions are depicted with the mean (dashed line) and the 5–95 and 25–75 interquantile ranges (shading).

## 3.2 Sensitivity analysis of the analog ensemble

AnEn's performance is sensitive to the number of analogs chosen from the historical data set (Figure 2). In the case of this study, 15–25 members produced ensemble mean forecasts of $O_3$ and $PM_{2.5}$ with the





highest correlations and forecasts of $PM_{2.5}$ with the lowest RMSEs. The lowest RMSEs from $O_3$ forecasts were achieved with 10–20 members. These sensitivities motivated our choice of 20 members for this particular study. Other studies will exhibit different sensitivities and might call for different measures of AnEn's performance, not simply RMSE and correlation.

5   The curves in Figure 3 result from two opposing trends. The more analogs employed, the more thoroughly we sample the statistical relationships between forecasts and observations in the training data. However, as each additional analog is included, the similarity between it and the current forecast decreases. Under typical circumstances, the shorter the training period (i.e., the larger the percentage of the training data from which ensemble members are drawn), the less analogous are the chosen analogs,

10 which will normally lead to lower correlations and higher RMSEs.

   Another way to measure this sensitivity is to evaluate AnEn's performance as a function of training period with the number of analogs held constant (Figure 4). Longer training periods improve AnEn's performance. The degree of improvement depends on the variable and metric. Forecasts of $O_3$ are improved most.





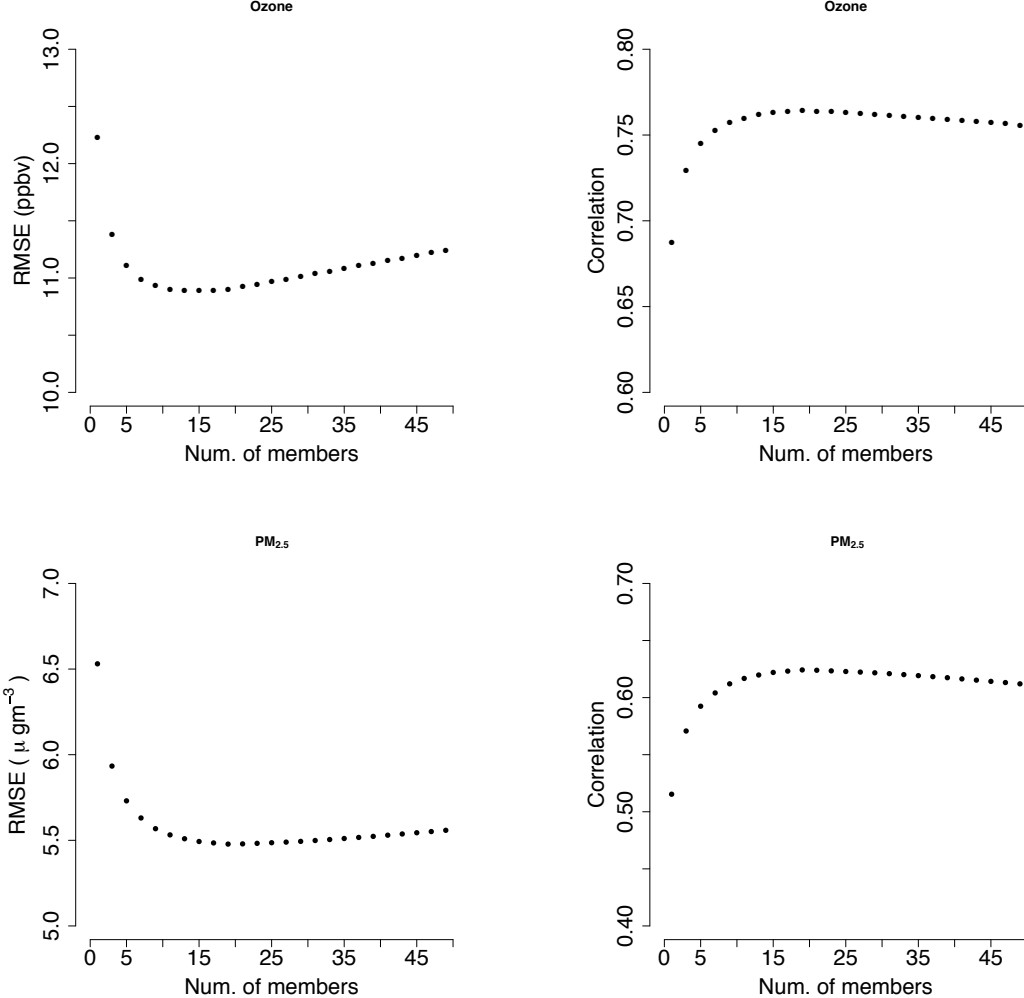

**Figure 3. Sensitivity of AnEn to number of ensemble members (i.e., analogs). AnEn forecasts' RMSE of a) O₃ (ppbv) and c) PM₂.₅**

**(µg m⁻³) and correlation of b) O₃ (0.0–1.0) and d) PM₂.₅ (0.0–1.0) vs. number of ensemble members (i.e., analogous forecasts selected**

5 **from the training archive). Calculations are averages over all lead times and sites during the periods of study described in the text.**

**Note the different ranges among *y* axes.**



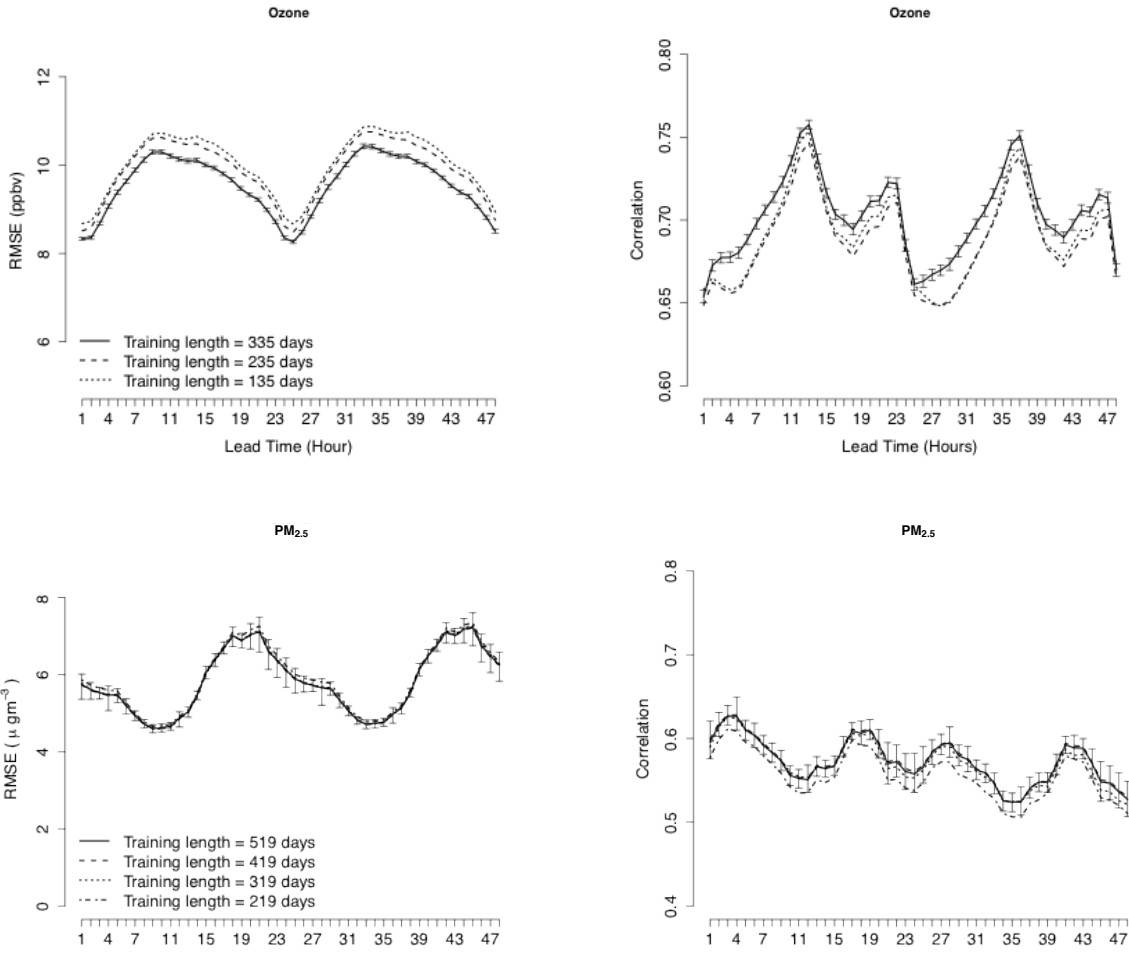

**Figure 4. Sensitivity of AnEn to length of the training period. AnEn forecasts' RMSE of a) O$_3$ (ppbv) and c) PM$_{2.5}$ (µg m$^{-3}$) and**

5    **correlation of b) O3 (0.0–1.0) and d) PM2.5 (0.0–1.0) vs. lead time (h) for the training periods shown by line style. Calculations are**

**averages over all sites during the periods of study described in the text. The number of analogs is 20 in every case. The vertical bars**

**indicate the 95% confidence intervals computed with bootstrapping.**



The weights $w_i$ for each of the five variables are determined independently for each observing site according to an algorithm that minimized continuous ranked probability score (CRPS) over the optimization periods of May 2015 ($O_3$) and November 2015 ($PM_{2.5}$)—for details on analog predictor optimization strategies, see the paper by Junk et al. (2015). The optimization periods do not overlap with

the period over which the performance metrics have been calculated. Weights can have values in the 0.0– 1.0 range with increments of 0.1, and the weights of $O_3$ and $PM_{2.5}$ CMAQ predictors are set to a minimum of 0.4 for the prediction of $O_3$ and $PM_{2.5}$, respectively. Figure 5 shows the distribution of the weights for each predictor for both $O_3$ (left) and $PM_{2.5}$ (right). For the prediction of $O_3$, the predictors are wind speed (WSPD, m s$^{-1}$) and direction (WDIR, degrees from N) at 10 m AGL, air temperature (T2M, ˚C) at 2 m

AGL, cloud fraction (CF), and ground-level concentrations of $O_3$ (ppbv). For $PM_{2.5}$, the analog predictors are T2M, WSPD, WDIR, specific humidity (Q, g kg$^{-1}$), and surface $PM_{2.5}$ (µg m$^{-3}$).

The distributions show the weights' variability across the stations. Both $O_3$ and $PM_{2.5}$ are weighted high for their respective predictions, as expected. For $O_3$, T2M, WSPD, and WDIR are weighted similarly, while the distribution of the weights for CF corresponds to the lowest values. For $PM_{2.5}$, T2M

has the second-highest median of its weight distribution followed by WSPD, WDIR, and then Q.



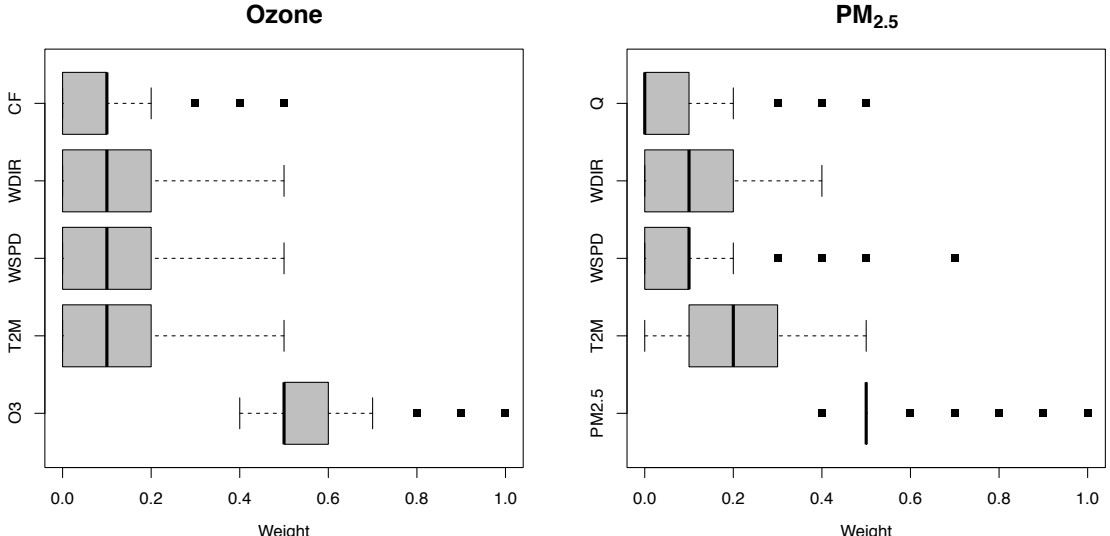

**Figure 5.** **Distribution of analog predictor weights across the available stations for O$_3$ (left) and PM$_{2.5}$ (right). The brown boxes indicate the 25–75 interquantile range, the black line within the box is the median, the squares are the outliers, and the vertical black lines at the edge of the dashed lines are the minimum and maximum excluding the outliers. O$_3$ predictors include wind speed (WSPD, m s$^{-1}$) and direction (WDIR, degrees from N) at 10 m AGL, air temperature (T2M, ˚C) at 2 m AGL, cloud fraction (CF), and ground-level concentrations of O$_3$ (ppbv). The PM$_{2.5}$ predictors used in this study are T2M, WSPD, WDIR, specific humidity (Q, g/kg), and surface PM$_{2.5}$ (μg m$^{-3}$).**

### 3.3 Deterministic predictions

When deterministic predictions are evaluated over all forecasts and observations in our study, AnEn's

10    mean forecasts of O$_3$ and PM$_{2.5}$ are dramatically superior to the raw forecasts from CMAQ (Figure 6).





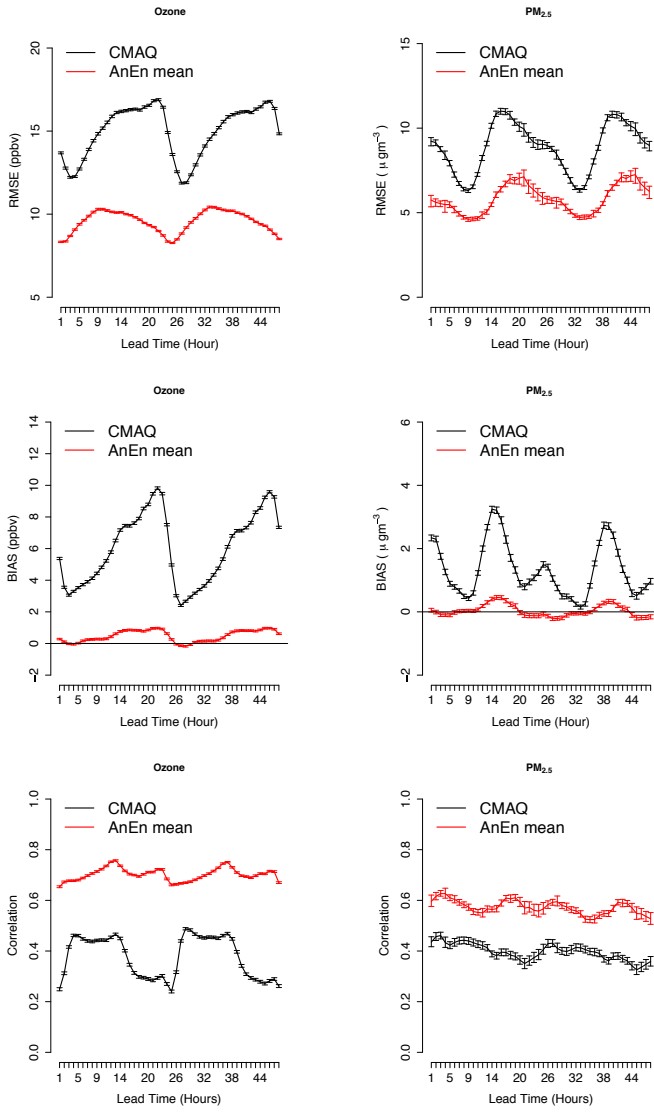

**Figure 6.** RMSE (top), bias (middle), and correlation (bottom), of O$_3$ (left) and PM$_{2.5}$ (right), vs. lead time in forecasts from CMAQ

5 (black) and AnEn mean (red). Calculations are averages over all sites during the periods of study described in the text.

The AnEn significantly improves CMAQ's raw prediction by reducing RMSE by approximately 35% ($O_3$) and 30% ($PM_{2.5}$), bias by 90% ($O_3$) and 95% ($PM_{2.5}$), and by improving the correlation by 50% ($O_3$ and $PM_{2.5}$). The metrics in Figure 6 vary with lead time because there is diurnal variation in CMAQ's

skill—which in turn affects AnEn's performance—and because observations are distributed across several time zones.

The AnEn significantly improves CMAQ estimates, which have been used to generate it. As shown in other applications (e.g., Delle Monache et al., 2011; Djalalova et al., 2015; Huang et al. 2017), it reduces both systematic and unsystematic errors, while significantly improving the correlation with

observations. In principal, given that the analog ensemble estimates are based on past observations, the AnEn mean should provide unbiased estimates. However, the residual bias after the AnEn correction (middle panels in Figure 6), is likely due by the fact that the training dataset is finite, which does not guarantee that the distribution of the observations that are the foundation for AnEn fully samples the predicted PDF.

Figure 7 and Figure **8** show the spatial distribution of the AnEn improvements (%) over CMAQ in RMSE and correlation for both $O_3$ and $PM_{2.5}$, computed independently at each available observation site and over the verification period. For $O_3$, AnEn consistently reduces CMAQ RMSE (Figure 7, top panel) with values 5−60%, with similar improvements to correlation (Figure **8**, top panel), although not as pronounced as to RMSE. AnEn improves CMAQ across different land uses, topographical



complexities, and geographic regions, resulting in a capability that can be considered for real-time

forecasting in operational centers. Similar results are observed for PM$_{2.5}$.

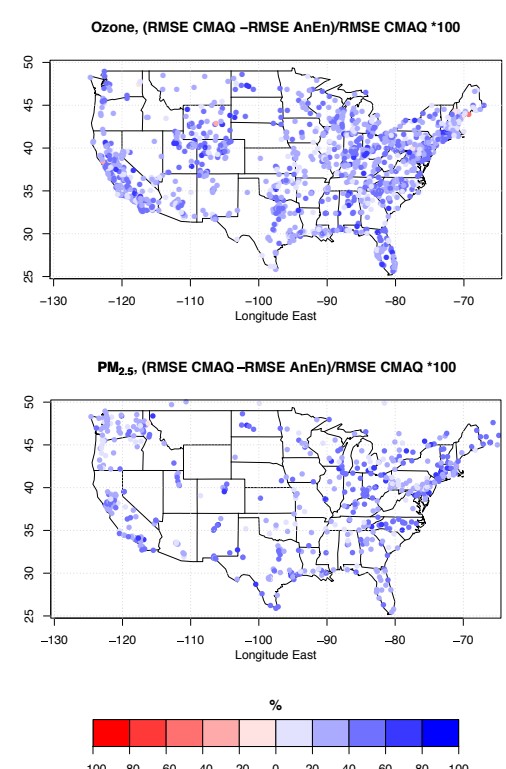

Figure 7. Improvement (% in color) to RMSE of forecasts of O$_3$ (top) and PM$_{2.5}$ (bottom) from AnEn vs. CMAQ.

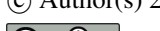


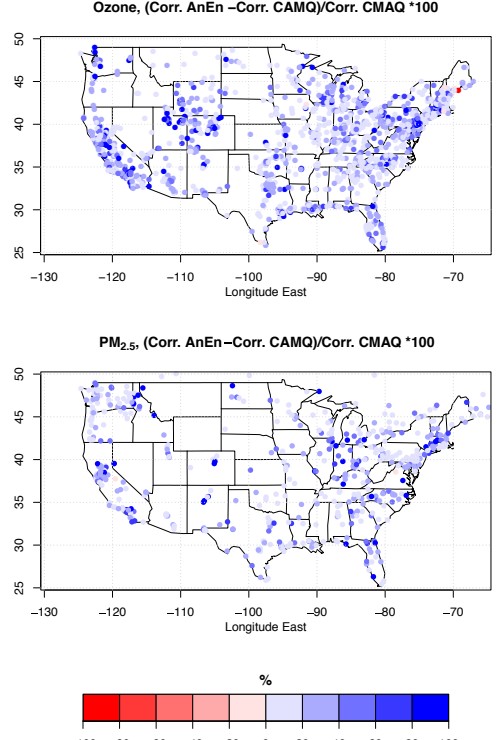

**Figure** 8. **Improvement (% in color) to correlation of forecasts of a) O$_3$ and b) PM$_{2.5}$ from AnEn vs. CMAQ.**

## 5   3.4  Probabilistic predictions

In this section, several attributes of AnEn and PeEn probabilistic predictions are evaluated to assess their performances for AQ forecasting. These include the match between the observed and predicted cumulative PDF, reliability, resolution, statistical consistency, and an analysis of the spread-skill relationships (Jolliffe and Stephenson, 2003; Wilks, 2006).





### 3.4.1 Observed vs. predicted cumulative distribution

The Continuous ranked probability score (CRPS) is computed to assess the closeness between observed

and predicted PDFs, by comparing the full ensemble distribution with the observations, where both

prediction and observations are represented as cumulative distribution functions, or CDF (Carney and

5   Cunningham, 2006). It corresponds to the mean absolute error of deterministic predictions, and it has the

same unit as the forecast variable. The more the ensemble-derived CDF is sharp and centered on the

corresponding observation, the lower the CRPS is. Zero is a perfect CRPS.

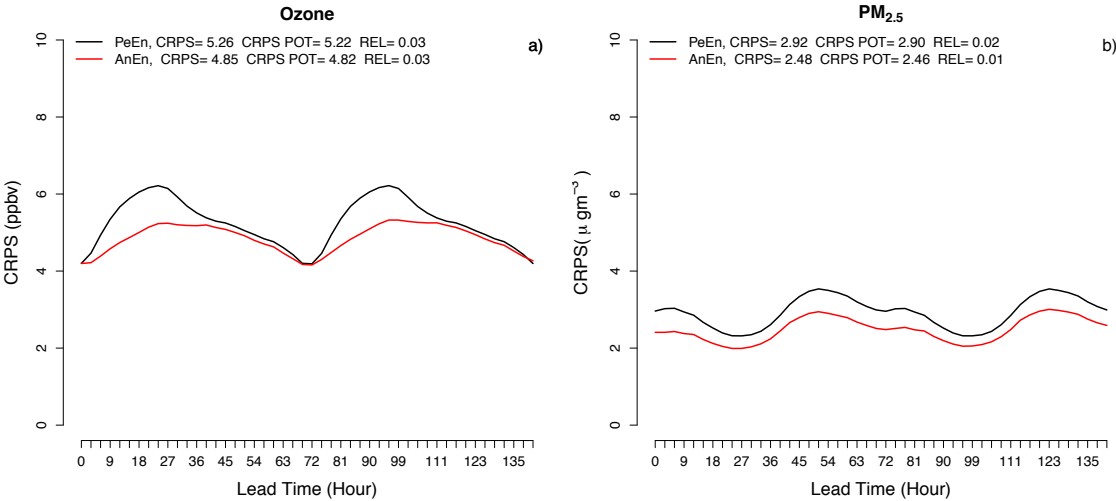

**Figure 9. Continuous ranked probability score (CRPS) vs. lead time from forecasts of a) O3 and b) PM2.5 by AnEn (red) and**
10   **PeEn (black).**

As shown in **Figure 9**, AnEn has a better (i.e., lower) CRPS than PeEn for most of the lead times of $O_3$

predictions, and all lead times of $PM_{2.5}$, indicating an overall better predictive probabilistic skill. The

distinct diurnal cycle in each CRPS series is not surprising, given the diurnal cycle in forecast error

(Figure 6). The better CRPS of AnEn results from better resolution, an important attribute of probabilistic

predictions. The reliability of the two systems is very similar. Reliability and resolution are discussed

next.

### 5  3.4.2  Reliability

An ensemble is reliable when its forecast probability matches the observed relative frequency (i.e., the

rate of occurrence) of a certain condition over a deep observational record. For instance, a reliable

ensemble will predict a 7% probability of ground-level $O_3$ concentration exceeding a regulatory threshold

at some point during a 24-h period if historically on 7% of days with similar conditions that threshold was

10  exceeded. For a given condition (e.g., $O_3 > 50$ ppb), plotting forecast probabilities from a perfectly

reliable model vs. observed relative frequencies will result in a 1:1 diagonal line on a reliability diagram

(Jolliffe and Stephenson, 2003; Wilks, 2006). However, the CRPS reliability component provides an

indication of overall reliability, i.e., not tied to a particular condition. For the latter, as shown in Figure

9 and indicated by "REL," the reliability of AnEn and PeEn are very similar, and close to the perfect

15  value of zero, indicating that both probabilistic prediction systems would provide an end-user with

information that is not misleading (i.e., that would not lead to over- or under-confidence in the forecast).

### 3.4.3  Resolution

The potential CRPS ("POT" in **Figure 9**) includes both uncertainty and resolution (Hersbach, 2000). The

uncertainty term is an attribute relative to the observations only, and is the same across different prediction

20  systems. However, the resolution quantifies the forecasts' ability to predict when an event occurs or not.



Probability forecasts with perfect resolution are 100% on occasions when the event occurs and 0% when the event does not occur. The CRPS POT values reported in **Figure 9** show that AnEn has better resolution than PeEn. This is because AnEn forecasts are designed to capture errors in the current raw prediction, whereas PeEn includes the most recent 20 observations of $O_3$ or $PM_{2.5}$ at the same hour of day as the

forecast valid time, which may not sample well the observation corresponding to the current prediction.

### 3.4.4 Statistical consistency

If an ensemble is statistically consistent, its members' forecasts are statistically indistinguishable from observations (Anderson, 1996). If this condition is satisfied, when ranking an observation against the corresponding ensemble forecasts, the observation falls with equal probability in any of the ranks. Over

a sufficient number of cases, when rank frequencies are plotted the resultant rank histogram is statistically flat if an ensemble is perfectly consistent (Anderson, 1996; Hamill, 2001; Talagrand et al., 1997). Rank histograms of forecasts from AnEn and PeEn are close to flat (Figure **10**). There is a slight high bias to both ensembles—shorter bars on the right of the histograms indicate that observations that fall among the higher predicted values are less common than those that fall among low values. The

ensembles are also slightly under-dispersive—the U shape at the tops of the bars indicate that observations that fall within the envelope of the ensembles' spreads are less common than those that fall outside the envelope, either lower than the lowest forecast value from a member or higher than the



highest forecast value.

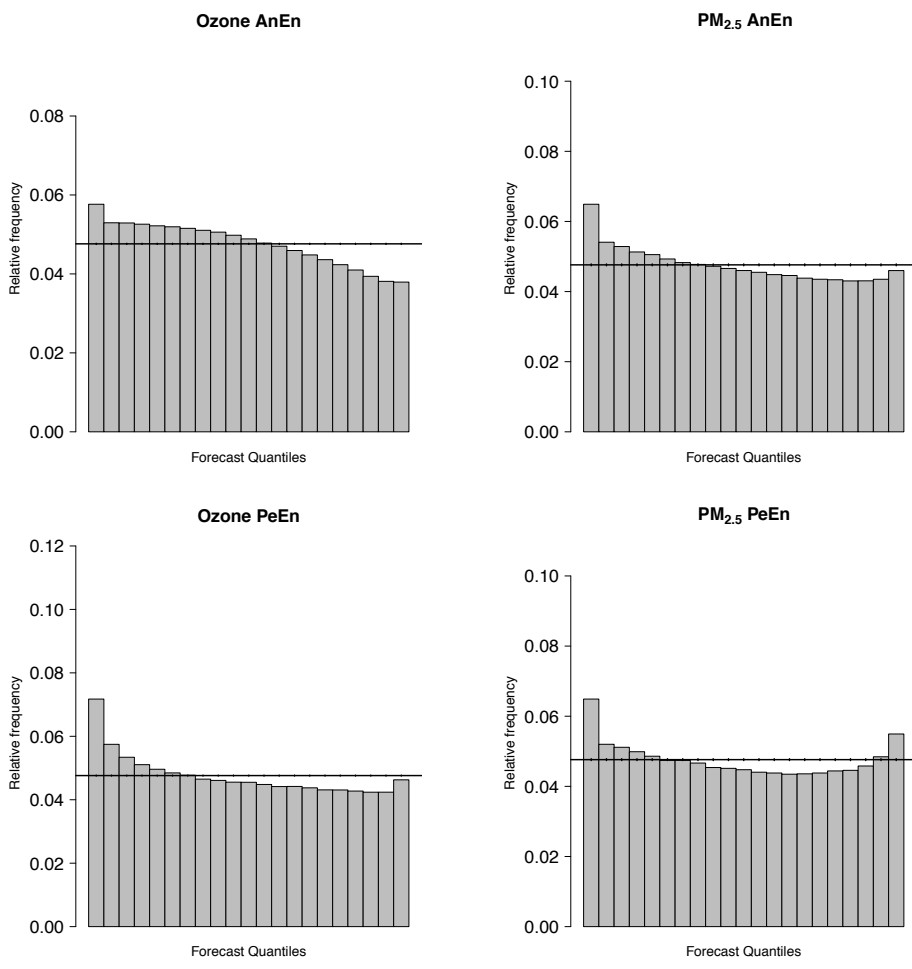

**Figure 10. Rank histograms of forecasts of O₃ (left) and PM₂.₅ (right) from AnEn (top) and from the persistence ensemble (bottom).**



### 3.4.5 Spread-skill relationship

One way to assess an ensemble system ability to quantify the prediction uncertainty is by relating the

spread among individual ensemble members' forecasts to the skill of their mean forecast, which is

referred to as the spread-skill relationship ( Delle Monache et al., 2013; van den Dool, 1989).  There are

various ways to measure this relationship.  Talagrand (1997) reasoned that a statistically consistent

ensemble's average standard deviation (a measure of spread) should match the RMSE of its mean

forecast.  Hopson (2014) provided insightful commentary on the topic.

We find that, indeed, standard deviation and RMSE correspond quite well on average (across the

different observational sites), especially for forecasts of $PM_{2.5}$ (Figure 11).  The correspondence is

extremely robust over the 48-h forecast period, depending much more strongly on the diurnal cycle than

on lead time.  AnEn does not share PeEn's tendency to be under-dispersive when forecasts of $O_3$ are

less accurate, but forecasts of $PM_{2.5}$ from both models are similarly under-dispersive.

One can also assess spread-skill relationship by examining model error vs. spread after the latter

is separated into bins that are subsets of the dataset's full range of spread.  We find that forecasts from

AnEn exhibit a strong spread-skill relationship according to this measure, as do forecasts from PeEn

(Figure 12).  Both ensembles are slightly under-dispersive when spread is small and slightly over-

dispersive when spread is large (evinced by the slopes < 1:1 in Figure 12), which is consistent with the

U-shaped rank histograms in Figure **10**.  The conditional bias displayed in Figure 12 is relatively minor,



however, and it might decrease with a larger archive of training cases (Delle Monache et al., 2013).

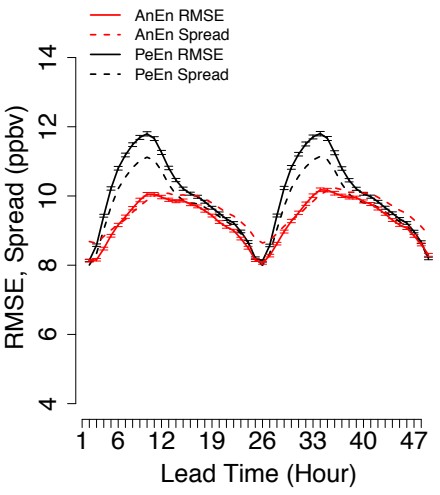
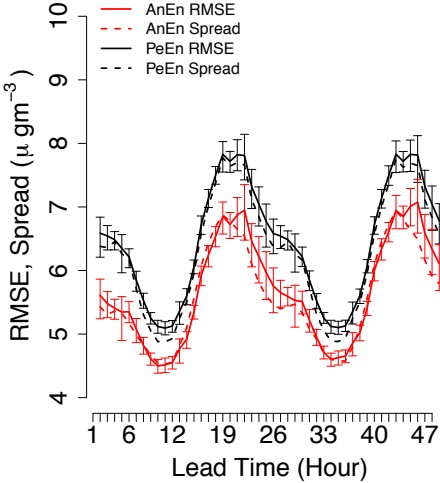

**Figure 11. Dispersion diagrams of forecasts of O$_3$ (ppbv, left) and PM$_{2.5}$ (µg m$^{-3}$, right) from the AnEn (red) and persistence ensemble (black). RMSE of the ensemble mean is solid, spread is dashed. Vertical bars span the 95% bootstrap confidence intervals.**

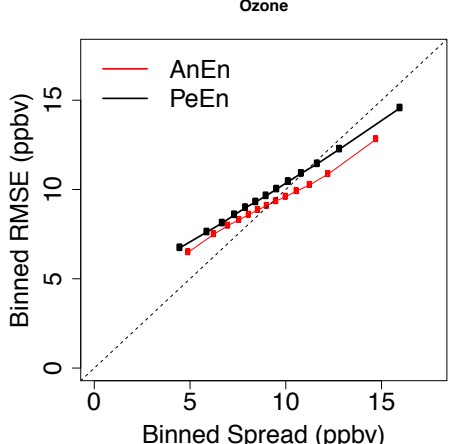
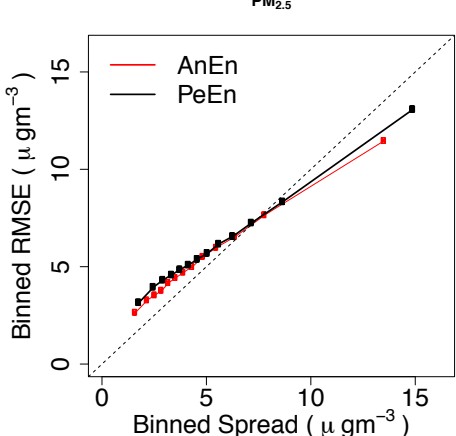





**Figure 12. RMSE vs. spread of forecasts of O$_3$ (ppbv, left) and PM$_{2.5}$ (µg m$^{-3}$, right) from the AnEn (red) and persistence ensemble (black) calculated over equal-size bins (dots).**

## 4 Summary

Conventional air-quality predictions are contaminated by uncertainties stemming from several sources, including initial conditions, emission, numerical approximations, and the simulation (or lack thereof) of physical and chemical processes. The ability to estimate these uncertainties in real time enhances decision-making to protect the public from poor air quality.

In this study, for the first time the analog ensemble (AnEn) technique has been applied to generate deterministic and probabilistic predictions of O$_3$ and PM$_{2.5}$. The available datasets span the period from 1 July 2014 through 30 September 2015 (456 days) for predictions of O$_3$ and 1 July 2014 through 29 February 2016 (608 days) for predictions of PM$_{2.5}$. The verification periods to assess the performances of the predictive systems are June through September 2015 for O$_3$ and December 2015 through February 2016 for PM$_{2.5}$. The analysis has been performed with 1945 and 458 stations for O$_3$ and PM$_{2.5}$, respectively, across the conterminous U.S. and southern Canada. The main findings are the following:

- The AnEn significantly improves the skill of deterministic predictions by reducing the errors of the deterministic model used to generate it, the Community Multiscale Air Quality (CMAQ), while increasing its correlation with the observations. For example, AnEn's root mean square error is lower than CMAQ's by roughly 35% and 30% for O$_3$ and PM$_{2.5}$, respectively.

- AnEn produces a probabilistic prediction which is statistically consistent, reliable, and sharp. It quantifies the uncertainty of the underlying prediction, which could contribute to an increased ability to protect the public health.

- An analog ensemble can be generated for existing real-time air-quality forecast systems with very

small additional computational cost in real-time. However, an analog ensemble does require an archive of historical simulations from a deterministic model and observations of the quantity to be predicted, which can be built offline.

The results reported herein can be further improved with a longer training dataset (which would require additional computational resources), by extending existing training datasets to consider neighboring

locations while searching analogs (at no additional computational cost), and by exploring more predictors for the analog-matching metric.

## 5 Code availability

The analog ensemble's code can be shared for research purposes upon request to the corresponding author.

## 6 Data availability

The datasets used in this study can be made available upon request to the corresponding author.



## 7  Author contributions

L. Delle Monache proposed the implementation of AnEn for air-quality predictions, designed the experiments and analysis, and coordinated the research team. He wrote the manuscript outline, the results, discussion, and conclusion sections, and contributed significantly to other sections of the manuscript.

5  S. Alessandrini contributed to the experiments' design and ran the analog ensemble algorithm to generate the results presented in the manuscript. He also generated initial versions of all the figures and edited the manuscript.

I. Djalalova performed the quality control of the measurements. She contributed to interpretation of results and edited the manuscript.

10  J. Wilczak contributed to the experiments' design, contributed to interpretation of results, and edited the manuscript.

J. Knievel wrote the abstract, the introduction, the explanation of the analog ensemble, and contributed to the results section. He edited the manuscript and many of the figures.

## 8  Competing interests

15  The authors declare that they have no conflicts of interest.

## 9  Acknowledgements

This work was supported by NASA Earth Science Division Applied Science Program (Grant # NNX15AH03G). We thank Jianping Huang (NCEP) for providing the CMAQ and Airnow data sets used in this analysis.





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
