# Peer review of "Air Quality Predictions with an Analog Ensemble"

_Atmospheric Chemistry and Physics, 2017_

## Referee Comment (RC1) · Anonymous Referee #1 · 23 May 2018

"The paper is yet another application of the technique already presented by the authors in several other instances, there is no new scientific value in the current manuscript, and the paper barely suites GMD. Many concepts are given for granted; several are the example of the imprecise use of language. The literature selection seems incomplete considering only weather forecast and air quality as examples, US papers mainly, and the author works. All literature on atmospheric dispersion that has preceded the air quality is neglected as if there is a specificity in the application case of techniques. The classification of multi-model ensemble is awkward (multi-physics, multi-model?) while there are classifications accepted that could be instrumental to the authors. The paper has a potential but not for ACP in my view."

This was my assessment at the "short report" level, but I am afraid by point of view has

not changed.

The paper is poor, it is just an application of a method used for weather prediction, solar power and now air quality forecasting. The authors clearly state it stems from Delle Monache et al. (2013, DM13), improved by the method of Alesandrinni(2015) and so much so that they do not even care about explaining the methods and present the novelty that relates to this paper. Then there is a publication in AE from almost the same authors referred to as "PM2.5 analog forecast and Kalman filter post-processing for the Community Multiscale Air Quality (CMAQ) model" that also refers to Delle Monache 2013 where the analog definition is also used and that already dealt with AQ. I see nothing new here compared to those publications and original that is worth publication in ACP. ACP aims at scientific novelty and originality since model developments and applications have long been confined to GMD.

The editorial style is that of an internal report, with reference to other publications for the details and rushing to the results. Sure the results are good but what is the surprise here? that a method that works when applied to dense networks of data (in space and time) works also when the variable is not called temperature but solar radiation or ozone or PM2.5? This is not serious in my view and this paper does not fit ACP at all. It did not at the beginning, I pointed it out, I gave an opportunity to intervene and make clear what is the scientific relevance, nothing has changed, I am sorry but I must reject it.

Accepting this publication not only would give a false sense of value to the authors, which was indeed in DM13, but it is not present here, but also would take away value from a large number of scientifically original and valuable publications that are present in the journal to date. To which very demanding reviews were presented that challenge the scientific standpoints at their very essence.

This is an application of a method that has the value of yet another application. The value of demonstrating that something that was proven in the past works also for this

case, as if one were to publish papers on the laws of gravity showing that they work to a couch, a lorry, a cow, a planet, an asteroid or the pen that sits in front of me on my desk.

The authors are strongly encouraged to rewrite it and submit it to GMD, which is as prestigious and rigorous as ACP and fitting much better the content of this paper.

Do re-write it however since the English is a bit strange at times and many concepts are rushed over like for example:

 c The disputable idea that operational AQ forecast prevents deaths and societal costs. In my view, the planning does more than the operational forecasting in that respect

 c Ensembles of many different kinds are discussed generically as ensembles but never presented for what they are and their differences

 c Many omissions in terms of ensemble applications are present thus giving a false sense of completeness to the paper content

These are minor issues compared to the lack of originality and scientific relevance but yet they will become important if the authors opt of GMD.

---

## Referee Comment (RC2) · Anonymous Referee #2 · 8 Jun 2018

The paper presents the use of an analog ensemble (AnEn) technique to improve air quality forecasts. The AnEn is applied to outputs of a numerical model for air quality, specifically on O3 and PM2.5. It relies on past observations and the corresponding forecasts to draw an ensemble of analog situations. The improvements in the forecast are assessed by means of different scores, and compared to references.

The presented method seems to improve the forecast in different ways and it might be relevant for this kind of applications. However, as I'm not working with air quality, I cannot judge how the method stands against other model output statistics or statistical postprocessing techniques in that context. The paper also gives the impression that the authors do not come from this field, as the provided specific literature on that topic is rather poor. There is no mention of other M.O.S. approaches, whereas there should be

some. Moreover, I'm not certain about the novelty of this study compared to previous works of the authors.

The whole manuscript is not so well written and is often difficult to follow. It should be rewritten in a more fluent way, and it should better describe the methods used. The frequent use of "we" and "our" is inadequate. A substantial work on the language should be done through the whole manuscript.

The predictors used in the method are introduced very late, in the middle of the results, while they should be introduced earlier. Moreover, there is no justification for the choice of these predictors. Please better explain the choice of the predictors and the method itself.

How does the method perform for extremes ? I suspect that the peaks, which are the most relevant to forecast, might not be well covered by the ensemble due to the very limited size of the observations that can be used as analogs. Additionally, how would the derived deterministic time series (the mean) work for more extreme values?

Specific comments:

- The calibration / verification periods should be clearly explained in the beginning of the manuscript, and the independence of the verification period specifically detailed. It is not clear which results are provided for the calibration or verification period. Is Fig. 2 in the verification period ?

- The number of references are unbalanced. There are too many for some points (e.g. P2L14-17), while some assertions have no reference.

- P3L4-5: not clear

- P3L16-17: issue with the ()

- P5L5: geopotential height at 500hPa is generally a predictor in analog methods, not a predictand (likely the same for 2-m dew point)

- P6L19: t=1: what is the unit ? days ?

- P7L2: specify the section where the number of analogs is optimized

- P10L8: Figure 3

- P11L2: sensitivities ?

- Figure 2: On the verification period? Is it the best reproduced days, or are they representative of the skill of the method?

- Figures 3, 4, . . .: a) b) c) d) not present on the figures

- P14L3: May and Nov or May to Nov ?

- P17L11-14: + they might not sample the observation archive uniformly

- P20L2-5: Not clear how you process it

- P20L12: They are = or very close in some cases!

- P21L18-19: not clear

- P22L2: a slightly better resolution, but not much. . .

- P24L16-18: Not clear which spread you are taking about

- The summary should not contain the details of the periods, but the results should be more discussed.

---

## Author Comment (AC1) · 6 Sep 2018

**Responses to Reviewer Comments on "Air Quality Predictions with an Analog Ensemble"**

**We thank the editor for coordinating the review of our manuscript, and the two anonymous reviewers for a thorough review of our manuscript. Please find our point by point responses to all the comments below. All of the reviewer's comments appear in regular font and our responses appear in the bold font.**

**Response to Reviewer #1**

Comment: "The paper is yet another application of the technique already presented by the authors in several other instances, there is no new scientific value in the current manuscript, and the paper barely suites GMD. Many concepts are given for granted; several are the example of the imprecise use of language. The literature selection seems incomplete considering only weather forecast and air quality as examples, US papers mainly, and the author works. All literature on atmospheric dispersion that has preceded the air quality is neglected as if there is a specificity in the application case of techniques. The classification of multi-model ensemble is awkward (multi-physics, multi-model?) while there are classifications accepted that could be instrumental to the authors. The paper has a potential but not for ACP in my view." This was my assessment at the "short report" level, but I am afraid by point of view has not changed. The paper is poor, it is just an application of a method used for weather prediction, solar power and now air quality forecasting. The authors clearly state it stems from Delle Monache et al. (2013, DM13), improved by the method of Alesandrinni(2015) and so much so that they do not even care about explaining the methods and present the novelty that relates to this paper. Then there is a publication in AE from almost the same authors referred to as "PM2.5 analog forecast and Kalman filter post-processing for the Community Multiscale Air Quality (CMAQ) model" that also refers to Delle Monache 2013 where the analog definition is also used and that already dealt with AQ. I see nothing new here compared to those publications and original that is worth publication in ACP. ACP aims at scientific novelty and originality since model developments and applications have long been confined to GMD. The editorial style is that of an internal report, with reference to other publications for the details and rushing to the results. Sure the results are good but what is the surprise here? that a method that works when applied to dense networks of data (in space and time) works also when the variable is not called temperature but solar radiation or ozone or PM2.5? This is not serious in my view and this paper does not fit ACP at all. It did not at the beginning, I pointed it out, I gave an opportunity to intervene and make clear what is the scientific relevance, nothing has changed, I am sorry but I must reject it. Accepting this publication not only would give a false sense of value to the authors, which was indeed in DM13, but it is not present here, but also would take away value from a large number of scientifically original and valuable publications that are present in the journal to date. To which very demanding reviews were presented that challenge the scientific standpoints at their very essence. This is an application of a method that has the value of yet another application. The value of demonstrating that something that was proven in the past works also for this case, as if one were to publish papers on the laws of gravity showing that they work to a couch, a lorry, a cow, a planet, an asteroid or the pen that sits in front of me on my desk. The authors are strongly encouraged to rewrite it and submit it to GMD, which is as prestigious and rigorous as ACP and fitting much better the content of this paper. Do re-write it however since the English is a bit strange at times and many concepts are rushed over like for example: The disputable idea that operational AQ forecast prevents deaths and societal costs. In my view, the planning does more

than the operational forecasting in that respect. Ensembles of many different kinds are discussed generically as ensembles but never presented for what they are and their differences. Many omissions in terms of ensemble applications are present thus giving a false sense of completeness to the paper content These are minor issues compared to the lack of originality and scientific relevance but yet they will become important if the authors opt of GMD.

**Reply: We strongly disagree with the reviewer's assessment that the paper has no new scientific value. The reviewer has raised five concerns, i.e., a simple replication of the technique, novelty of the work, citing relevant literature, paper writing, and relevance to ACP. Here, we respond to these concerns one by one.**

**Simple replication of the analog ensemble technique****: Yes, we have employed the analog ensemble technique in a variety of applications but, contrary to the reviewer's belief, the previous applications do not guarantee that the technique will work for air quality application – a statement that is true for any method, which is the reason why in science the generality of a new algorithm is never guaranteed and needs to be tested. The predictive skill of numerical models for weather parameters (e.g., wind and temperature) may vary significantly when compared to the predictive skill of models for quality variables (e.g., ozone and $PM_{2.5}$), and that can significantly affect the performance of any given postprocessing method. For example, the extension of analog ensemble application from one area to another (e.g., weather to air quality) requires careful selection of the predictors that best identify the similar (analogous) atmospheric conditions in the past.**

**For air quality, the predictors have to be selected in such a way that they are able to (1) identify air pollution episodes of similar magnitude in the past, and (2) identify the meteorological and chemical conditions leading to similar past air pollution episodes. Following these two criteria, we selected $O_3$, $PM_{2.5}$, 10-m wind speed and direction, 2-m air temperature, 2-m specific humidity, and cloud cover as the predictor variables in our implementation of analog ensemble for air quality. The rationale for selecting these variables as predictors is now described in the revised manuscript and reproduced here for reference:**

**"The rational for selecting the aforementioned air quality and meteorological variables as predictor variables is as follows. $O_3$ and $PM_{2.5}$ allow us to identify pollution episodes of similar magnitude in the past. Temperature plays a vital role in several processes relevant to air quality including atmospheric chemical kinetics, biogenic emissions, and mixing. The wind speed and wind direction allow us to ensure that similar transport pathways contributed to the analogous air pollution episodes in the past. Humidity is selected for its key role in the formation and destruction of both $O_3$ and $PM_{2.5}$. Water vapor ($H_2O$) in conjunction with $O_3$ photolysis is the main source of hydroxyl (OH) radical, which in turn initiates photochemical production of $O_3$ through oxidation of different volatile organic compounds (VOCS). In the case of $PM_{2.5}$, humidity determines the aerosol water content, which is important for secondary aerosol formation. Cloud cover determines the amount of solar radiation available for atmospheric photochemical reactions that produce both $O_3$ and $PM_{2.5}$. In summary, the predictors are strategically selected in such a way that they are not only able to identify the pollution episodes of similar magnitude in the past but also identify the meteorological and chemical conditions leading to similar air pollution episodes."**

**Because of the research required to identify the suitability of a method for a given application, we believe it is naive to compare different applications of analog ensemble with**

the application of universal law of gravity that simply requires replacement of masses of the bodies involved. Furthermore, research also allows us to challenge the universally accepted laws. For example, even following the line of thought of the reviewer, the Newton's law of gravity and motion were found to be not accurate enough to deal with very strong gravitational fields or to describe with extreme precision the orbit of Mercury around the Sun. Scientists discovered that when trying to apply the Newton's laws to objects other than a couch or a cow.

Novelty of the work: To the best of our knowledge, we are proposing for the first time a novel approach to generate probabilistic predictions for air quality, which is based on a significant shift in paradigm with respect to traditional ensemble methods: i.e., rather than running a numerical model with several different configurations to create the ensemble members, we run the air quality model in real time only once, and then generate the necessary uncertainty quantification by inference from the training data set. Additionally, a strategic selection of the predictors is required for using the analog ensemble method in air quality applications (see the response above for details). The selection of these predictors with their corresponding weights (as explained in Section 2.3 and 3.2) contributes further to the novelty of this work. It is worth noticing that Djalalova et al. (2015) did not touched the subject of probabilistic predictions as in this proposed work, it was focused only on $PM_{2.5}$ (here we analyze the performance of the proposed method on both ozone and $PM_{2.5}$), and it involved the combination of analog-based deterministic methods with the Kalman filter (the latter is not part of the current manuscript).

Citing relevant literature: We did cite several papers on transport and dispersion modeling as requested by this Reviewer in his preliminary comment (i.e., we cite among others Galmarini et al., 2001; Galmarini et al. 2004; Kioutsioukis and Galmarini et al. 2014; Potempski et al. 2008; Potempski and Galmarini, 2009; Solazzo et al. 2012). Here, we focus on reducing biases and quantifying uncertainty in the air quality forecasts produced by the three-dimensional Eulerian chemistry transport model (CTM). Therefore, we discuss potential sources of uncertainties in CMAQ and the relevant literature attempting to reduce biases and errors in CTM forecasts. We believe that citing our previous work is quite relevant and fully consistent with ACP guidelines, to provide a reader with the background information on where and how the analog ensemble has been used previously.

Paper Writing: We apologize for the imprecise use of language. The manuscript has been revised throughout to improve clarity and proof-read by a native English speaker. We agree with the reviewer that planning is important to mitigate health impacts of air pollution but operational air quality forecasts are as important because air quality managers cannot plan anything until they know about the forthcoming air pollution episodes. Several sections of the manuscript including Introduction, Prediction System and Data, and results have been revised to reflect this sentiment.

Relevance to ACP: Here, we have neither developed nor described a numerical model and/or model component, which is a key requirement for the GMD. Rather, we provide robust evidence that the analog ensemble technique is capable of reducing errors and biases in air quality predictions, and to generate reliable and calibrated probabilistic predictions. ACP

has previously published articles that focused on reducing errors and biases in air quality predictions either using chemical data assimilation over both the US and Europe, or a dynamical ensemble in China (e.g., Saide et al., 2013; Flemming et al., 2017; Kioutsioukis et al., 2016; Potempski and Galmarini, 2009; Hu et al., 2017). Therefore, we strongly believe that our paper is suitable for publication in ACP rather than GMD.

References

Flemming, J., Benedetti, A., Inness, A., Engelen, R. J., Jones, L., Huijnen, V., Remy, S., Parrington, M., Suttie, M., Bozzo, A., Peuch, V.-H., Akritidis, D., and Katragkou, E.: The CAMS interim Reanalysis of Carbon Monoxide, Ozone and Aerosol for 2003–2015, Atmos. Chem. Phys., 17, 1945-1983, https://doi.org/10.5194/acp-17-1945-2017, 2017.

Saide, P. E., Carmichael, G. R., Liu, Z., Schwartz, C. S., Lin, H. C., da Silva, A. M., and Hyer, E.: Aerosol optical depth assimilation for a size-resolved sectional model: impacts of observationally constrained, multi-wavelength and fine mode retrievals on regional scale analyses and forecasts, Atmos. Chem. Phys., 13, 10425-10444, https://doi.org/10.5194/acp-13-10425-2013, 2013.

Hu, J., Li, X., Huang, L., Ying, Q., Zhang, Q., Zhao, B., Wang, S., and Zhang, H.: Ensemble prediction of air quality using the WRF/CMAQ model system for health effect studies in China, Atmos. Chem. Phys., 17, 13103-13118, https://doi.org/10.5194/acp-17-13103-2017, 2017.

---

## Author Comment (AC2) · 6 Sep 2018

**Response to Reviewer #2**

The paper presents the use of an analog ensemble (AnEn) technique to improve air quality forecasts. The AnEn is applied to outputs of a numerical model for air quality, specifically on O3 and PM2.5. It relies on past observations and the corresponding forecasts to draw an ensemble of analog situations. The improvements in the forecast are assessed by means of different scores, and compared to references.

The presented method seems to improve the forecast in different ways and it might be relevant for this kind of applications. However, as I'm not working with air quality, I cannot judge how the method stands against other model output statistics or statistical postprocessing techniques in that context. The paper also gives the impression that the authors do not come from this field, as the provided specific literature on that topic is rather poor. There is no mention of other M.O.S. approaches, whereas there should be some. Moreover, I'm not certain about the novelty of this study compared to previous works of the authors.

**Reply: To the best of our knowledge, we are proposing for the first time a novel approach to generate probabilistic predictions for air quality, which is based on a significant shift in paradigm with respect to traditional ensemble methods: i.e., rather than running a numerical model with several different configurations to create the ensemble members, we run the air quality model in real time only once, and then generate the necessary uncertainty quantification by inference from the training data set. Additionally, a strategic selection of the predictors is required for using the analog ensemble method in air quality applications. Specifically, the predictors have to be selected in such a way that they are able to (1) identify air pollution episodes of similar magnitude in the past, and (2) identify the meteorological and chemical conditions leading to similar past air pollution episodes. Following these two criteria, we selected $O_3$, $PM_{2.5}$, 10-m wind speed and direction, 2-m air temperature, 2-m specific humidity, and cloud cover as the predictor variables in our implementation of analog ensemble for air quality. The rationale for selecting these variables as predictors is now described in the revised manuscript and reproduced in response to your comment on introduction of predictors in the manuscript.**

**In this study, we compare the performance of analog ensemble against the Persistence ensemble and show that the analog ensemble performs better (section 3.1). We appreciate the reviewer suggestion of comparing our method against other approaches such as MOS but we would prefer to perform a comprehensive comparison of our methods with the others such as MOS using a common dataset for all the methods rather than comparing the results from different studies that focus on different regions with different objectives and model configuration. We have cited many papers on the ensemble modeling in the Introduction Section and also added more references on transport and dispersion modeling following suggestion from Reviewer #1 (e.g., we cite Galmarini et al., 2001; Galmarini et al. 2004; Kioutsioukis and Galmarini et al. 2014; Potempski et al. 2008; Potempski and Galmarini, 2009; Solazzo et al. 2012).**

The whole manuscript is not so well written and is often difficult to follow. It should be rewritten in a more fluent way, and it should better describe the methods used. The frequent use of "we" and "our" is inadequate. A substantial work on the language should be done through the whole manuscript.

**Reply: We apologize for the imprecise use of language here. The manuscript has been revised thoroughly and proof read by a native English speaker.**

The predictors used in the method are introduced very late, in the middle of the results, while they should be introduced earlier. Moreover, there is no justification for the choice of these predictors. Please better explain the choice of the predictors and the method itself.

**Reply: We introduced the predictors in Section 2.3 along with the description of CMAQ modeling system. The revised manuscript also provides a justification for the use of predictors. The new text is reproduced here for your ready reference. "The rational for selecting the aforementioned air quality and meteorological variables as predictor variables is as follows. $O_3$ and $PM_{2.5}$ allow us to identify pollution episodes of similar magnitude in the past. Temperature plays a vital role in several processes relevant to air quality including atmospheric chemical kinetics, biogenic emissions, and mixing. The wind speed and wind direction allow us to ensure that similar transport pathways contributed to the analogous air pollution episodes in the past. Humidity is selected for its key role in the formation and destruction of both $O_3$ and $PM_{2.5}$. Water vapor ($H_2O$) in conjunction with $O_3$ photolysis is the main source of hydroxyl (OH) radical, which in turn initiates photochemical production of $O_3$ through oxidation of different volatile organic compounds (VOCS). In the case of $PM_{2.5}$, humidity determines the aerosol water content, which is important for secondary aerosol formation. Cloud cover determines the amount of solar radiation available for atmospheric photochemical reactions that produce both $O_3$ and $PM_{2.5}$. In summary, the predictors are strategically selected in such a way that they are not only able to identify the pollution episodes of similar magnitude in the past but also identify the meteorological and chemical conditions leading to similar air pollution episodes in the past."**
**Regarding the method description, our previous paper (Delle Monache et al., 2013) already presents a step-by-step description of the basic technique and we have reproduced the necessary details here. Here, we focus on describing the ways in which the application of AnEn to AQ differ from previous applications rather than repeating the information from the published literature.**

How does the method perform for extremes ? I suspect that the peaks, which are the most relevant to forecast, might not be well covered by the ensemble due to the very limited size of the observations that can be used as analogs. Additionally, how would the derived deterministic time series (the mean) work for more extreme values?
**Reply: This is an excellent question, and we agree with the reviewer that an analysis of extreme events, which has now been added, significantly enriches the paper. To understand the performance of AnEn for extreme events, we computed the bias, RMSE, and correlation coefficient for both the CMAQ forecasts and AnEn derived deterministic time series of ozone and $PM_{2.5}$ using only the observations above the 95% quantile, computed independently at each lead time and observation location. The estimated bias, RMSE, and correlation coefficient for extreme events are shown in Figures R1, R2, and R3, respectively. A lower RMSE and higher correlation coefficient of AnEn derived deterministic time series for both ozone and $PM_{2.5}$ at all the lead times shows that AnEn outperforms CMAQ for the extreme events. However, the bias in AnEn is higher than CMAQ raw forecasts for extreme events of**

**PM₂.₅** **mainly because of substantial reduction in the number of available quality analogs when we consider only extreme events. A lower RMSE even at the lead times where AnEn bias is higher indicates that AnEn compensate the latter by reducing the random errors, i.e., Centered Root Mean Square Error (CRMSE) in the forecasts. Our future work will focus on a bias correction technique to reduce the AnEn bias for the extreme events. This information has been included in the revised manuscript.**

[Figure]

**Figure R1: Estimated bias in CMAQ forecasts and AnEn derived deterministic forecasts of ozone (left panel) and PM₂.₅ (right panel) for the extreme events that are identified as observations above 95% quantile of the distribution.**

[Figure]

**Figure R2: Same as R1 but for the RMSE.**

[Figure]

**Figure R3: Same as R1 but for the correlation coefficient.**

Specific comments: -

The calibration / verification periods should be clearly explained in the beginning of the manuscript, and the independence of the verification period specifically detailed. It is not clear which results are provided for the calibration or verification period. Is Fig. 2 in the verification period ?

**Reply: The verification and training periods are now defined right before section 3.1. The following text has been added to the manuscript. "The verification periods are selected as 1 June to 30 September 2015 for O₃ because O₃ is a major air quality problem during summertime; and 1 December 2015 to 29 February 2016 for PM₂.₅ because PM₂.₅ pollution is higher during wintertime. Consequently, the training periods for O₃ and PM₂.₅ are selected as 1 July 2014 to 31 May 2015, and 1 July 2014 to 30 November 2015, respectively." Yes, Fig. 2 represents the verification period.**

- The number of references are unbalanced. There are too many for some points (e.g. P2L14-17), while some assertions have no reference.

**Reply: We agree that we have a large number of references at this line but we think it is really important to acknowledge the previous research conducted on this subject.**

- P3L4-5: not clear
**Reply: This sentence has now been changed to:**

**"A *well resolved* ensemble is one that provides a probability close to 100% on occasions when an event (e.g., ozone above 100 ppb) occurs and forecast close to 0% when the event**

**does not occur. I.e., it is specific from case to case about what conditions will and will not occur.**

- P3L16-17: issue with the ()
**Reply: Thanks for pointing this out. Correct ( ) are placed now.**

- P5L5: geopotential height at 500hPa is generally is generally a predictor in analog methods, not a predictand (likely the same for 2-m dew point)
**We agree with the Reviewer that Geopotential height at 500 hPa has been used as a predictor in past applications of AnEn, and it is an important variable to consider when generating weather forecasts. In fact, through and ridges in its field are a proxy for indicating area of instability and/or underling low/high pressure systems. It can be relevant for air quality as well, but likely not as much as the predictors we have selected. We now recognize that the list of predictors we selected may not be exhaustive (at the beginning of the second paragraph of section 2.3).**

- P6L19: t=1: what is the unit ? days ?
**Reply: We apologize for the confusion and this has now been corrected; 1 represents hours.**

- P7L2: specify the section where the number of analogs is optimized

**Reply: The section number is mentioned now.**

- P10L8: Figure 3

**Reply: Yes, we meant Figure 3. Corrected.**

- P11L2: sensitivities ?

**Reply: Changed to sensitivity.**

- Figure 2: On the verification period? Is it the best reproduced days, or are they representative of the skill of the method?
**The verification period has been chosen independently of the model's performance. We chose a winter period for $PM_{2.5}$ and a summer period for $O_3$ because these are the season when these pollutants have higher concentrations.**

- Figures 3, 4, . . .: a) b) c) d) not present on the figures
**Reply: Corrected.**

- P14L3: May and Nov or May to Nov ?

**Reply: This are May 2015 and Nov 2015 and not May to Nov 2015.**

- P17L11-14: + they might not sample the observation archive uniformly

**We agree with the reviewer. As already mentioned, these testing periods are those most significant because they usually include high pollution episodes for the two pollutants considered.**

- P20L2-5: Not clear how you process it

**Reply: The sentence has been broken down into two parts now to improve readability. The CPRS is equivalent to the mean absolute error of deterministic predictions relative to the observations.**

- P20L12: They are = or very close in some cases!

**Reply: Yes, the CPRS for AnEn and PeEn are similar at lead time of around 72 h and 135 h. That is why we wrote that AnEn has a better (i.e., lower) CRPS than PeEn for "most of the lead times" of O₃ predictions and not for "all the lead times".**

- P21L18-19: not clear
**We added some text to better clarify the point.**

- P22L2: a slightly better resolution, but not much. . .
**Reply: The improvement of the AnEn over PeEn in resolution are about 10% and 15% for ozone and PM2.5 respectively. We specified this in the paper.**

- P24L16-18: Not clear which spread you are taking about
**Reply: We specified that the spread is defined as the standard deviation of the members about the ensemble mean.**

- The summary should not contain the details of the periods, but the results should be more discussed.
**Reply: We believe that it is helpful to remind the readers of the period for which the study is conducted in the Summary section of the manuscript.**